# Perioperative Risk Prediction in Major Gynaecological Oncology Surgery: A National Diagnostic Survey of UK Clinical Practice

**DOI:** 10.3390/diagnostics15131723

**Published:** 2025-07-06

**Authors:** Lusine Sevinyan, Anil Tailor, Pradeep Prabhu, Peter Williams, Melanie Flint, Thumuluru Kavitha Madhuri

**Affiliations:** 1School of Applied Sciences, University of Brighton, Brighton BN2 4GJ, UK; 2Department of Gynaecological Oncology, Royal Surrey NHS Foundation Trust, Guildford GU2 7XX, UK; 3Department of Anaesthetics, Royal Surrey NHS Foundation Trust, Guildford GU2 7XX, UK; 4Department of Mathematics, University of Surrey, Guildford GU2 7XH, UK; 5Venture Development Center, University of Massachusetts, Boston, MA 02125, USA

**Keywords:** gynaecological oncology, risk prediction, perioperative diagnostics, major surgery, morbidity, mortality, P-POSSUM, ACS NSQIP, surgical outcomes

## Abstract

**Background**: Gynaecological oncology (GO) surgery involves a wide range of procedures, from minor diagnostic interventions to highly complex cytoreductive operations. Accurate perioperative diagnostics—particularly in major surgery—are critical to optimise patient care, predict morbidity, and facilitate shared decision-making. This study aimed to evaluate current practices in perioperative risk assessment amongst UK GO specialists, focusing on the use, perception, and applicability of diagnostic risk prediction tools. **Methods**: A national multicentre survey was distributed via the British Gynaecological Cancer Society (BGCS) to consultants, trainees, and nurse specialists. The questionnaire examined clinician familiarity with and use of existing tools such as POSSUM, P-POSSUM, and ACS NSQIP, as well as perceived reliability and areas for improvement. **Results**: Fifty-four clinicians responded, two-thirds of whom were consultant gynaecological oncologists. While 51.9% used morbidity prediction tools selectively, only 7.4% used them routinely for all major surgeries. The most common models were P-POSSUM (39.6%) and ACS NSQIP (25%), though over 20% did not use any formal tool. Despite this, 80% of respondents expressed a desire for more accurate, GO-specific models. **Conclusions**: This study reveals a gap between available perioperative diagnostics and real-world clinical use in GO surgical planning. There is an urgent need for validated, user-friendly, and GO-specific risk prediction tools—particularly for high-risk, complex surgical cases. Further research should focus on prospective validation of tools such as ACS NSQIP and their integration into routine practice to improve outcomes in gynaecological oncology.

## 1. Introduction

Gynaecological cancers remain a significant global health challenge, accounting for 7.3% of new cancer cases and 7.0% of cancer-related deaths worldwide in 2022, as reported by the most recent Global Cancer Statistics (GLOBOCAN 2022) [1]. Cervical and uterine cancers are identified as the most frequently diagnosed malignancies in women, ranking second and sixth globally, respectively. However, cervical cancer is no longer the second most common malignancy in many high-income countries, where effective screening and HPV vaccination programmes have significantly reduced its incidence. In contrast, the burden remains disproportionately high in low- and middle-income countries. Despite the widespread implementation of HPV vaccination and the World Health Organization’s goal to eliminate cervical cancer by 2030, age-standardised incidence rates for both cervical and uterine cancers continue to rise in many regions, highlighting persistent global disparities [1].

This increasing disease burden has driven a parallel rise in surgical interventions, particularly technically challenging minimally invasive surgeries (MIS) [2] in the rising population of obese patients [3] and patients with multiple comorbidities, due to an ageing population [4]. Major surgical procedures in gynaecological oncology (GO) surgery are associated with considerable perioperative morbidity influenced by both patient- and surgery-specific factors [5,6].

The accurate prediction of perioperative complications is essential in GO, as major postoperative complications can delay or prevent the delivery of adjuvant treatment, a critical component in the management of many gynaecological malignancies. Such delays can compromise optimal treatment delivery and adversely affect disease-free intervals and overall survival [7,8,9]. Emerging evidence also suggests that postoperative complications may directly influence oncological outcomes by increasing the risk of cancer relapse, thereby shortening disease-free survival, which serves as a key indicator of patient quality of life—even if overall survival is not significantly impacted [10,11]. Preoperative functional status and overall health have been consistently identified as key predictors of postoperative morbidity, with poorer baseline health linked to both increased incidence and severity of complications following major GO surgery [12,13]. These complications—such as anastomotic leak, fistula formation, infections, thromboembolism, and renal, cardiac and other complications—are associated with prolonged hospitalisation, higher rates of readmission, and increased healthcare utilisation and costs [13,14]. Increased longevity coupled with comorbidities causes a substantial clinical burden and with the growing number of high-risk surgical candidates, the ability to accurately stratify perioperative risk through multidisciplinary assessment is critical. Inadequate risk stratification may result in avoidable morbidity and even mortality as well as poor risk management and breakdown of communication between patients, their support networks and healthcare providers [15,16]. Risk stratification supports personalised treatment planning, improves the quality of shared decision-making, and enables the implementation of proactive strategies to mitigate perioperative risk, ultimately aiming to reduce morbidity and enhance overall patient quality of life [17,18].

Accurate preoperative diagnostics—especially risk prediction tools—are therefore essential to improve perioperative outcomes, facilitate shared decision-making, manage expectations around postoperative care and recovery, personalise surgical planning and ultimately a patient’s quality of life (QoL).

In this context, perioperative risk prediction models serve as critical diagnostic tools. They enhance the informed consent process and support compliance with professional guidance, including the General Medical Council’s emphasis on shared decision-making and individualised risk communication [19].

The Royal College of Obstetricians and Gynaecologists (RCOG) similarly emphasises the importance of incorporating patient-specific comorbidities, medication history, and pre-existing medical conditions into surgical risk discussions [20,21]. Evidence suggests that objective, visualised, and personalised risk assessments improve patient understanding, expectations, and engagement in decision-making [22].

Despite the availability of various diagnostic scoring systems, there is currently no universally adopted risk prediction model specifically validated for GO procedures. Anaesthetists often use the American Society of Anesthesiologists (ASA) Physical Status Classification [23], and oncologists may refer to performance status scales such as Eastern Cooperative Oncology Group (ECOG) [24] or Karnofsky [25] to guide treatment decisions, particularly chemotherapy, although these tools reflect functional status rather than surgical risk.

More direct perioperative diagnostics include the Physiologic and Operative Severity Score for the Enumeration of Mortality and Morbidity (POSSUM) and Portsmouth POSSUM (P-POSSUM) scores [26,27], Surgical Outcome Risk Tool (SORT) [28], Acute Physiology and Chronic Health Evaluation II (APACHE II) [29], Surgical Apgar Score (SAS) [30], Surgical Risk Scale [31], Donati Surgical Risk Score [32] and Charlson Comorbidity Index (CCI) [33], age-adjusted CCI [34], *MySurgeryRisk* [35]. However, many of these models were developed outside of GO and may not fully reflect the nuances of gynaecological cancer surgery.

Among the more comprehensive options, the American College of Surgeons National Surgical Quality Improvement Program (ACS NSQIP) risk calculator [36,37] stands out as a more comprehensive and informative tool. It incorporates 19 preoperative patient variables including the specific procedure the patient is due to undergo to provide customised, patient-specific predictions for 11 different postoperative outcomes within 30 days of surgery as well as the length of hospital stay [38]. Nonetheless, its diagnostic utility in the GO setting remains under-evaluated.

As part of an ongoing investigation into perioperative diagnostics in gynaecological oncology, we conducted a national, multicentre survey titled “Perioperative Risk Prediction in Gynaecological Oncology”.

This study aimed to explore the current use, perceived reliability, and clinical integration of perioperative risk prediction tools in GO across the United Kingdom (UK). Specifically, we sought to evaluate clinicians’ familiarity with and utilisation of existing diagnostic models such as POSSUM, P-POSSUM, and ACS NSQIP, and to identify opportunities for improving perioperative diagnostics tailored to GO surgical patients.

Perioperative risk prediction tools do serve a diagnostic function by identifying patients at higher risk of morbidity and mortality, facilitating shared decision-making, optimising resource allocation, and improving overall surgical outcomes. Yet, despite their potential, these tools are not consistently adopted in GO clinical practice, and few are validated specifically for this patient population.

Our study addresses this gap by surveying UK-based GO specialists to characterise real-world diagnostic practices and to assess interest in refining or adopting more accurate, GO-specific tools. By doing so, we aim to inform future validation studies, encourage evidence-based adoption of diagnostic tools, and ultimately support precision surgery in gynaecological oncology.

## 2. Materials and Methods

### 2.1. Survey Development

The authors developed a cross-sectional survey which was subsequently peer-reviewed by the British Gynaecological Cancer Society (BGCS) committee (Table A1). The questionnaire was designed to assess current perioperative risk prediction practices among GO specialists, their perspectives on the effectiveness of existing risk calculators and their views on potential future developments. The survey was conducted using the SurveyMonkey^®^ online platform and was structured into eight sections. Each question was designed in a multiple-choice answer format, with optional open-ended feedback sections for additional qualitative responses. All survey questions were single-response items, except for Question 4, which permitted multiple selections to capture the use of more than one prediction algorithm.

### 2.2. Participants and Distribution

The survey targeted actively practicing GO specialists across the UK who were members of BGCS with valid email addresses. Eligible participants included:Consultant gynaecological oncologists based at tertiary cancer centres,Obstetrics and gynaecology (O&G) consultants as part of cancer units,Subspecialty and specialty trainees in GO based at tertiary cancer centres,Clinical nurse specialists (CNS) in GO based at tertiary cancer centres.

An anonymous online questionnaire was distributed via the BGCS membership forum, ensuring broad participation among UK GO specialists. Participation in the survey was voluntary. The invitation included information regarding the study’s rationale and contact details for further inquiries.

### 2.3. Data Analysis

Data were collected using the SurveyMonkey^®^ platform and exported into Microsoft Excel^®^ and SPSS v27 (IBM Corp., Armonk, NY, USA) for analysis. Descriptive statistics were calculated to summarise clinician demographics, frequency of tool use, perceived reliability, and awareness of specific risk prediction tools.

Categorical variables (e.g., use of specific tools, frequency of use, professional role, setting) were summarised using counts and percentages.

Ordinal responses (e.g., agreement with statements about reliability or perceived need) were reported as proportions, and Likert-type responses were grouped (e.g., “strongly agree” and “agree”) where appropriate for clarity.

Missing data were accounted for by adjusting denominators for each question.

Graphical representations (Figure 1, Figure 2, Figure 3, Figure 4 and Figure 5) were generated to visually demonstrate patterns in tool use, perceived reliability, and clinician familiarity with ACS NSQIP.

Open-ended qualitative responses were reviewed manually and summarised narratively to identify emerging themes or concerns.

No inferential statistics were performed due to the exploratory nature of the study and the sample size. However, results offer actionable insights and lay the groundwork for hypothesis-driven prospective validation studies.

## 3. Results

### 3.1. Demographics

A total of 54 respondents participated in the survey. The majority (65%) were consultant gynaecological oncologists working in gynaecological cancer centres, while the remaining respondents included O&G consultants, subspecialty and specialty trainees, and CNSs. A summary of demographic data from the first two survey questions is presented in Table 1.

### 3.2. Utilisation of Morbidity Prediction Algorithms

Table 2 summarises the responses regarding the use of morbidity and mortality (M&M) prediction algorithms in GO surgery. Each survey question, but one (Question 4) was structured to permit only one response option.

The findings indicate that the use of these algorithms is selective, with only 7.4% of specialists applying them to all patients undergoing major surgery. In contrast, 22.2% of respondents reported never using any risk prediction tools (Figure 1). Notably, none of the participants selected the option “I do not perform surgery,” confirming that all respondents were actively involved in surgical decision-making.

A total of 48 algorithm selections were recorded in response to Question 4, indicating that some respondents selected more than one algorithm. Percentages in Table 2 are based on these 48 total selections. When asked about specific risk prediction tools used in perioperative assessment (Question 4), 39.6% and 16.7% of respondents reported using POSSUM and P-POSSUM, respectively, while 25% utilised the ACS NSQIP calculator. Other tools mentioned under the “Other” category included SORT and Charlson Comorbidity Index (CCI) (Table 2, Figure 2).

### 3.3. Perceived Reliability of Risk Prediction Tools

In Question 5, participants were asked to assess the reliability of their chosen risk prediction tools for perioperative M&M.

Responses (Table 2, Figure 3) revealed that approximately 40% of specialists considered their current tool reliable, while over 80% of respondents acknowledged the need for a more accurate and tailored risk prediction model for GO surgical procedures (Figure 4).

### 3.4. Familiarity with ACS NSQIP

The survey also assessed clinicians’ awareness and usage of the ACS NSQIP calculator (Question 7). The findings (Figure 5) highlight limited familiarity with this tool among GO specialists in the UK. Over one-third of respondents had never heard of ACS NSQIP or were unaware of its applicability in GO surgery. Only approximately 30% of respondents had used it or had observed its use in clinical practice.

## 4. Discussion

The early detection and diagnostic precision of gynaecological malignancies have witnessed considerable advancements in recent decades. Innovations in imaging, biomarker discovery, and minimally invasive biopsy techniques have collectively enabled earlier identification of disease, more accurate staging, and better-targeted treatments, leading to significantly improved patient outcomes [39]. Nevertheless, in the modern era of gynaecological oncology (GO), the role of diagnostics must expand beyond the point of identifying disease. Instead, a more holistic approach is needed—one that includes tools for predicting surgical risk, postoperative morbidity, and long-term functional recovery impacting a patient’s QoL. This would represent a natural step forward in diagnostics, supporting the broader objective of precision medicine that aim to customise clinical interventions to the individual characteristics of each patient.

In this context, perioperative risk prediction tools serve a vital diagnostic function. These tools provide estimations of a patient’s likelihood of experiencing adverse outcomes based on a combination of physiological, surgical, and disease-specific variables. By enabling the delivery of more individualised care, they support both clinical decision-making and patient-centred communication. From a patient’s perspective, the provision of personalised and objective estimates of surgical risk can enhance understanding of the proposed intervention, inform consent discussions, and foster shared decision-making. For clinicians, these tools facilitate more accurate planning of perioperative care pathways, inform resource allocation, and help identify patients who may benefit from targeted interventions such as prehabilitation, enhanced recovery protocols, or intensive postoperative monitoring.

Risk stratification approaches are not uniform across medical disciplines. These practices vary significantly across medical specialties, reflecting the specific clinical priorities and contexts of each discipline. Anaesthetists commonly use the ASA Physical Status Classification System, which offers a subjective but widely accepted assessment of a patient’s baseline physiological reserve [23]. Oncologists, meanwhile, frequently rely on performance status measures, such as the Eastern Cooperative Oncology Group (ECOG) or the Karnofsky Performance Status (KPS), to assess a patient’s functional capacity and eligibility for systemic therapies [24,25]. While useful, these scales are not designed specifically to predict surgical outcomes. Surgeons, particularly those operating in high-risk specialties such as GO, need more comprehensive diagnostic tools that incorporate a wider array of patient and procedural variables.

Gynaecological oncology (GO) procedures include a wide range of surgical interventions, ranging from relatively minor procedures such as LLETZ (large loop excision of the transformation zone) to highly complex operations like cytoreductive debulking for advanced ovarian cancer. The relevance and necessity of perioperative risk prediction tools vary accordingly. While such tools may offer limited utility in low-risk, minor interventions, they become critically important in major abdominal procedures—particularly among high-risk patient populations, such as those who are elderly, obese, or have multiple comorbidities. [6,40,41,42] These high-complexity GO procedures often involve multivisceral surgeries that may extend beyond the reproductive organs to include the bowel, bladder, or upper abdomen. These procedures are often performed in patients with significant comorbidities, obesity, or age-related frailty—factors that independently elevate perioperative risk [3,4]. The increasing adoption of minimally invasive (MIS) techniques, including robotic-assisted surgery, within the field of GO has partially mitigated some of these risks by reducing surgical trauma and facilitating faster recovery [43]. Robotic systems improve surgical precision, enhance visualisation, and reduce surgeon fatigue [44], thereby expanding the operability window for patients previously considered too high-risk for open surgery, by minimising physiological stress and perioperative risk.

Given the complexity and variability of GO surgery, accurate risk assessment is essential. Personalised perioperative diagnostic measures allow clinicians to tailor anaesthetic plans, optimise comorbidities, and implement targeted intraoperative and postoperative strategies. These diagnostics become particularly important in cases of borderline operability, where surgical decisions must be weighed against the possibility of significant morbidity. Thus, risk prediction is not merely adjunctive but rather a foundational component of the diagnostic process in modern surgical oncology. This study explored the perspectives of practicing GO specialists regarding the use of perioperative risk prediction models and their integration into clinical practice. This is the first UK-wide survey focused exclusively on the perioperative diagnostic practices in GO, filling a critical evidence gap.

Our national survey of GO specialists revealed considerable variability in the adoption and use of perioperative risk prediction tools. Although several models exist—including the POSSUM, P-POSSUM, and the ACS NSQIP calculator—their use in clinical practice is inconsistent. Only 7.4% of respondents reported routinely using such tools for all patients, while 22.2% did not use any formal risk prediction tool. The most used models, POSSUM and P-POSSUM, require intraoperative data, limiting their utility for preoperative diagnostics.

The ACS NSQIP surgical risk calculator offers notable advantages by integrating 19 preoperative variables and procedure-specific inputs to estimate multiple postoperative outcomes, including serious complications, reoperations, and length of hospital stay [36,37]. Crucially, it does not rely on intraoperative data, allowing for its use during preoperative assessment and patient counselling. Despite these advantages, only 25% of clinicians reported using ACS NSQIP in their practice. This relatively low uptake may be attributable to a variety of factors presented in the literature. Many surgeons report relying primarily on clinical judgement and experience [45], especially in high-risk, complex fields like GO, where general risk models may not account for the unique multivisceral surgery and oncological variables involved. Tools such as ACS NSQIP, designed primarily for the general surgical population, are often viewed as insufficient for GO procedures, which can often involve multivisceral resections, extensive adhesiolysis, or cytoreductive techniques in patients who may present in either a pre-treated state with neo-adjuvant chemotherapy or those who are treatment-naïve with significant fluid shifts pre-operatively during primary debulking for ovarian cancer. This mismatch leads to doubts about how useful and accurate these tools are in this clinical setting [46,47,48]. Additionally, logistical constraints further limit the adoption of prediction models. Many models require manual data input, are not integrated into electronic health record systems, or are perceived as time-consuming—factors that limit their practicality in busy clinical environments [45,48,49]. Our survey also revealed a notable lack of awareness among UK GO specialists regarding the ACS NSQIP surgical risk calculator, with 37% of respondents indicating they had never heard of the tool and an additional 15% unaware of its potential applicability within the GO setting.

Notably, while only 15% of respondents felt that their current risk prediction tools were inadequate, more than 80% expressed a desire for improved models tailored to the GO population. This suggests a disconnect between clinical practice and the recognition of tool limitations. Traditional tools like POSSUM and P-POSSUM were not originally designed for GO patients and have demonstrated limited predictive accuracy in this population—particularly for those undergoing complex cytoreductive or multivisceral surgeries [50]. In contrast, our own retrospective pilot study demonstrated that ACS NSQIP provided more accurate risk stratification in GO patients compared to P-POSSUM [51]. These findings reinforce the need for models that are both validated and optimised for the unique characteristics of GO patients.

More broadly, perioperative risk prediction should be regarded as a distinct branch of diagnostics—one that complements traditional pathological and radiological investigations. Just as molecular diagnostics and imaging techniques have revolutionised the early detection of cervical and uterine cancers, perioperative diagnostics hold the potential to significantly improve surgical outcomes and the overall quality of cancer care.

For these tools to be adopted more widely, usability within the clinical workflow must be prioritised. Integration into electronic health records, incorporation into preoperative assessment pathways, and the development of intuitive user interfaces can facilitate routine use. Visual aids and individualised risk profiles generated from tools such as ACS NSQIP can be powerful educational and communication tools, especially when used in preoperative clinics or multidisciplinary team meetings.

Another compelling application of diagnostic risk prediction tools is in quality assurance and institutional benchmarking. In the UK, routine M&M meetings serve as critical platforms for continuous surgical quality improvement [52]. However, the lack of standardised risk prediction models in GO complicates inter-centre comparisons and fair auditing of surgical outcomes. Risk-adjusted outcome monitoring, underpinned by validated predictive diagnostics, could enable more meaningful institutional comparisons and drive quality improvement initiatives.

Our findings also resonate with international trends that highlight the importance of data-driven clinical pathways and outcome-based care. The global GO SOAR1 study found surgical morbidity rates exceeding 30% across both low- and high-income settings [53], underscoring the widespread need for enhanced perioperative strategies. In high-resource countries, diagnostics such as frailty assessments, cardiopulmonary exercise testing, and prehabilitation programmes are increasingly utilised to optimise surgical candidates. Risk prediction models can serve as integrative platforms, synthesising various assessments into actionable perioperative plans.

Clinician attitudes towards risk prediction tools significantly influence their adoption. Our survey results suggest that while many clinicians acknowledge the value of such tools, they may not routinely use them due to perceived barriers such as complexity, lack of time, or insufficient training. To address these challenges, a coordinated effort is urgently needed at both institutional and national levels. Educational initiatives, specialty-specific training, and formal endorsement through national guidelines for risk stratification in gynaecological oncology (GO) could substantially improve clinical uptake. Furthermore, healthcare systems and academic institutions should prioritise funding for large-scale, prospective validation studies to assess existing models and support the development of new tools designed specifically for GO populations.

At the same time, developers must focus on enhancing usability, clinical relevance, and predictive accuracy of these tools—particularly by tailoring them to the specific complexities of GO. This includes accounting for the unique aspects of GO procedures, which often involve multivisceral surgery and complex operative decision-making. Incorporating GO-specific variables—such as nutritional status of the patients, frailty, tumour histology, CA-125 levels, operative complexity scores, and extent of disease—could significantly enhance predictive accuracy, build clinician trust, and support more effective integration into surgical planning and shared decision-making.

Our survey offers valuable insights into the real-world utilisation of surgical risk assessment tools among GO specialists, revealing preferences for specific diagnostic tools and perceptions of their effectiveness. Unlike previous single-centre studies, this survey captures national perspectives and practice variation, identifying institutional differences and opportunities for standardisation. It also identifies discrepancies between clinician perceptions and objective predictive performance of these tools. However, several limitations must be acknowledged. First, the relatively small and UK-specific sample may limit the generalisability of the findings both within the UK and internationally. However, the culture within UK specialist training in GO, emphasising patient consultation and informed decision-making, as advocated by the GMC, encourages openness to improve patient outcomes and supports the value of conducting such surveys and future research. Caution should be exercised when extrapolating these results beyond the UK, as differences in healthcare infrastructure, clinical training, and perioperative practices may influence attitudes and usage patterns elsewhere. A larger, more diverse cohort—including participants from varied healthcare systems and geographic regions—would offer a broader and more representative understanding of global clinical practice and is currently underway. Second, inconsistencies in responses—such as participants indicating non-use of risk prediction tools while still answering related questions—suggest potential recall bias or variability in the interpretation of survey items. Furthermore, certain survey questions, such as *“There is a clinical need for a more accurate risk prediction tool for patients undergoing gynae-oncological surgical procedures”*, may have introduced response bias by implying the inadequacy of existing models. Additionally, although the survey explicitly stated in Question 3 that the focus was on the use of surgical risk tools in major gynaecological surgery, this may not have been sufficiently clear in the survey title or other questions. As a result, some respondents may have considered minor procedures, introducing further bias. Future studies should incorporate more neutral and precise wording to mitigate such effects. It is also relevant to consider the demographic composition of survey respondents. Approximately two-thirds were consultant gynaecological oncologists, while the remainder included general O&G consultants, trainees, and nurse specialists. The frequency with which each group performs major surgery may differ substantially, potentially affecting their perspectives on the relevance and utility of risk prediction tools. Stratifying responses by clinical role—and ideally by case volume or cancer type—could yield additional insights. Whilst this level of stratification was not practical in our dataset, future research could explore correlations between surgical caseload, subspecialisation, and diagnostic tool usage.

Despite these limitations, our findings provide a critical foundation for improving perioperative risk stratification in GO. To advance this effort, national and international stakeholders—including professional societies, funders, and research bodies—should prioritise the development, validation, and implementation of GO-specific risk prediction models. These efforts will be essential in ensuring that perioperative risk assessment becomes a standard component of preoperative diagnostics, thereby improving patient outcomes and supporting precision surgical care in GO.

## 5. Conclusions

This study identifies a significant gap in the diagnostic continuum of gynaecological oncology: the inconsistent use of validated perioperative risk prediction tools, despite their clear clinical advantages. As GO procedures grow in complexity and as patient populations become more comorbid and frailer, the role of predictive diagnostics in guiding surgical care becomes increasingly indispensable. Our data indicates that while tools like ACS NSQIP show considerable promise, their adoption remains suboptimal due to factors including lack of awareness, limited integration into clinical workflows, and absence of GO-specific validation.

There is a clear clinical demand for more robust, accurate, and user-friendly risk prediction tools that can support surgical decision-making, enhance communication with patients, and inform institutional benchmarking. Prospective validation studies and the development of GO-specific models are urgently required to bridge this gap. Embedding these diagnostics into routine clinical practice could transform the perioperative care landscape in GO, paving the way for more personalised, evidence-based interventions.

Ultimately, just as precision diagnostics have transformed the landscape of early cancer detection, the next frontier lies in deploying perioperative diagnostic tools to personalise surgical care and improve outcomes. As innovations in molecular, radiological, and pathological diagnostics continue to evolve, perioperative risk prediction must keep pace to ensure that surgical care remains safe, effective, and centred on the unique needs of each patient.

## Figures and Tables

**Figure 1 diagnostics-15-01723-f001:**
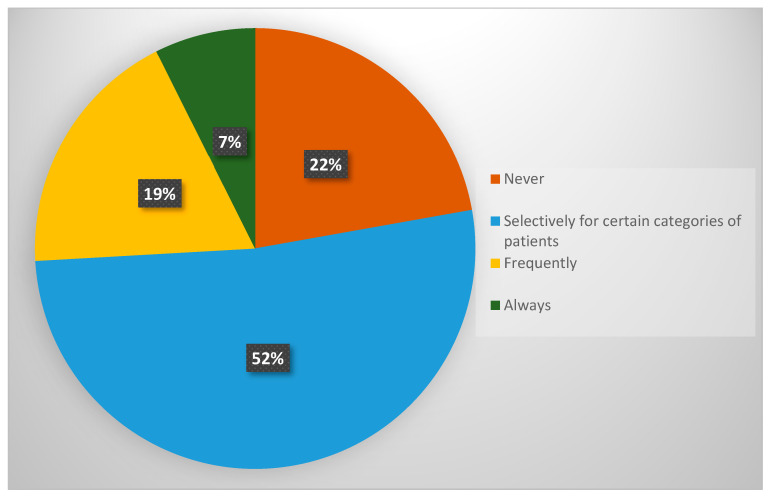
Utilisation of morbidity prediction algorithms prior to major surgery among GO specialists in the UK.

**Figure 2 diagnostics-15-01723-f002:**
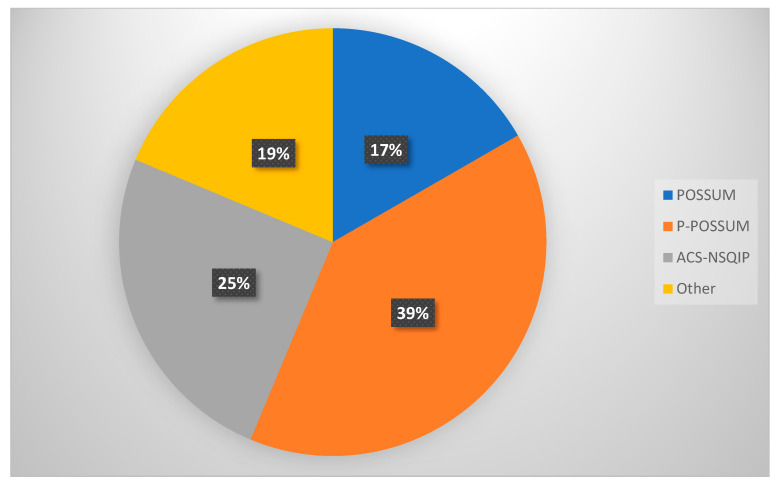
Risk prediction tools employed by GO specialists in the UK.

**Figure 3 diagnostics-15-01723-f003:**
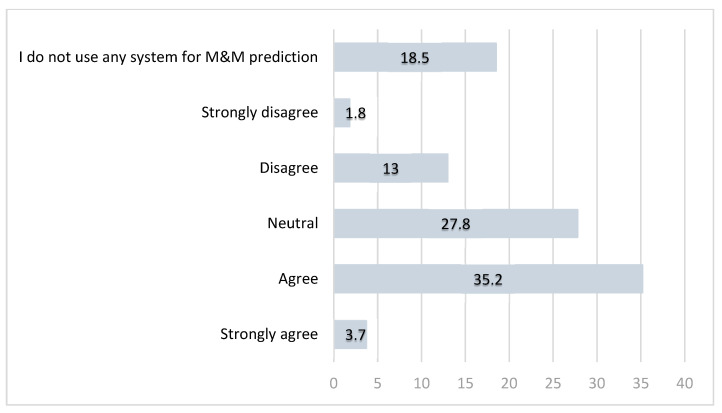
Opinions (in percentages) among GO specialists in the UK on whether current perioperative M&M risk prediction tools are reliable.

**Figure 4 diagnostics-15-01723-f004:**
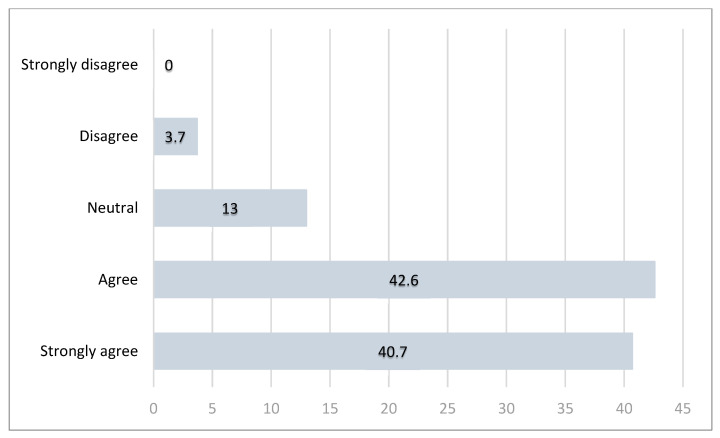
Responses (in percentages) distribution for Question 6—clinical need for a more accurate risk prediction tool in GO surgery.

**Figure 5 diagnostics-15-01723-f005:**
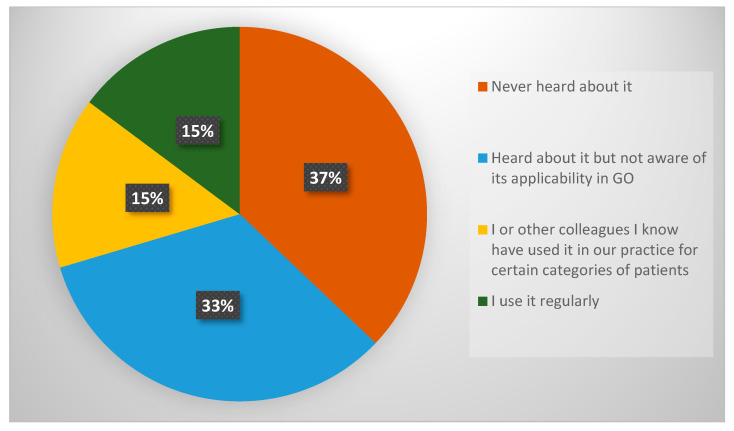
Response distribution for Question 7—familiarity with ACS NSQIP surgical risk calculator.

**Table 1 diagnostics-15-01723-t001:** Demographics of survey participants.

Variable	Number	%
Clinical role (Question 1) *		
Consultant GO	36	66.7
Subspecialty GO	5	9.3
Consultant O&G	7	13.0
Speciality trainee O&G	4	7.4
CNS	1	1.8
Other	1	1.8
Clinical setting (Question 2) *		
Gynaecological cancer centre	36	66.7
District general hospital	17	31.5
Other	1	1.8

* 100% response rate was recorded in these questions.

**Table 2 diagnostics-15-01723-t002:** Survey questions 3–5 with response variables.

Question	Response Variable	Number	%
*3. Do you use any algorithms to predict morbidity* *prior to their major surgery? **	Never	12	22.2
Selectively for certain categories of patients	28	51.9
Frequently	10	18.5
Always	4	7.4
I do not perform surgery	0	0
*4. What algorithms or risk scoring systems do you use? ***	POSSUM	8	16.7
P-POSSUM	19	39.6
ACS NSQIP	12	25.0
Other (please specify)	9	18.7
*5. The system I use is reliable for predicting peri-operative morbidity and* *mortality* * 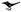 . **	Strongly agree	2	3.7
Agree	19	35.2
Neutral	15	27.8
Disagree	7	13.0
Strongly disagree	1	1.8
I do not use any system for predicting M&M	10	18.5

* 100% response rate was recorded in these questions. ** Percentages based on total algorithm selections (*n* = 48). Multiple responses per respondent were permitted.

## Data Availability

The original contributions presented in this study are included in the article. Further inquiries can be directed to the corresponding author.

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
