# Peer review of "Perioperative Risk Prediction in Major Gynaecological Oncology Surgery: A National Diagnostic Survey of UK Clinical Practice"

_diagnostics, 2025, doi:10.3390/diagnostics15131723_

Round 1
Reviewer 1 Report
Comments and Suggestions for Authors
Dear authors, the gynaecology-oncology-specific risk prediction issue is of great interest. It was a real pleasure to read this manuscript.
In the introduction please specify more clearly the main question of your study.
Do you consider the topic is relevant to the field?
In the methodology section you did not specify the Ethics Committee approval number.
Data analyses is too short and seems incomplete.
What does your research add to the subject area, compared to other published material?
The results are well presented and the discussions are well conducted.
More references are needed.
Author Response
Reply attached

Reviewer 2 Report
Comments and Suggestions for Authors
This is a well written paper worthy of publication after minor revision according to the following comments.
- Introduction, first paragraph: Cervical cancer is the second most frequently diagnosed malignancy in women worldwide, but no longer in high-income countries. This discrepancy between high- and low-income countries should be added to this paragraph.
-Surgical procedures in Gynaecological Oncology vary significantly, from minimal procedures, such as LLETZ, to major abdominal surgery, such as surgical debulking for stage III ovarian cancer. Preoperative risk prediction is of utmost importance before major surgery, especially in high-risk patients, including obese and elderly patients, whereas this may not be necessary for minor procedures. According to question 3, this study primarily focused on major surgery, but this was not clear in the other questions or in the title of the questionnaire and the title of the paper. As this may have influenced responses, the authors should discuss this potential source of bias. Furthermore, the authors should probably change the title of the paper accordingly.
-Two thirds of those who responded to the questionnaire were Consultants in Gynaecological Oncology, while the remaining one third were not. This might be an additional source of bias, as the former may conduct major surgery much more frequently than the latter. Hence, it would be interesting to know specifically how the subgroup of Consultants in Gynaecological Oncology responded to the questionnaire. Ideally, it would be interesting to stratify responses according to the number of major procedures performed annually by respondents, and according to the type of cancer (ovarian, endometrial etc.) and stage of disease.
Author Response
Dear reviewer, please find our response attached. We are very grateful for your insightful thoughts.

Reviewer 3 Report
Comments and Suggestions for Authors
Please find the comments attached.
Thank you!

Author Response

(The authors gave the same response as above.)
